# Corrosion Resistance Mechanism of Mica–Graphene/Epoxy Composite Coating in CO_2_-Cl^−^ System

**DOI:** 10.3390/ma15031194

**Published:** 2022-02-04

**Authors:** Shi-Dong Zhu, Yan-Peng Li, Hong-Wei Wang, Jin-Ling Li, An-Qing Fu, Gang Chen, Dong Ma, Xuan-Peng Li, Frank Cheng

**Affiliations:** 1School of Materials Science and Engineering, Xi’an Shiyou University, Xi’an 710065, China; lypxsy@126.com (Y.-P.L.); madong19990623@163.com (D.M.); 2The 8th Oil Production Plant, PetroChina Changqing Oilfield Company, Xi’an 710014, China; wanghw_cq@petrochina.com.cn; 3Shaanxi Oil and Gas Pollution Control Technology and Reservoir Protection Key Laboratory, College of Chemistry and Chemical Engineering, Xi’an Shiyou University, Xi’an 710065, China; lijinling@xsyu.edu.cn (J.-L.L.); gangchen@xsyu.edu.cn (G.C.); 4State Key Laboratory for Performance and Structure Safety of Petroleum Tubular Goods and Equipment Materials, CNPC Tubular Goods Research Institute, Xi’an 710077, China; fuanqing@cnpc.com.cn (A.-Q.F.); lixuanpeng127@163.com (X.-P.L.); 5Department of Mechanical and Manufacturing Engineering, University of Calgary, Calgary, AB T2N 1N4, Canada; fcheng@ucalgary.ca

**Keywords:** epoxy resin, the coating, corrosion resistance, graphene, mica

## Abstract

The working environment for tubing in oil and gas fields is becoming more and more serious due to the exploration of unconventional oil and gas resources, leading to the increasing need for a protective internal coating to be used in tubing. Therefore, a new mica–graphene/epoxy composite coating with different graphene contents (0.0, 0.2, 0.5, 0.7, and 1.0 wt.%) was prepared to improve the tubing resistance to a corrosive medium, an autoclave was used to simulate the working environment, and an electrochemical workstation assisted by three-electrodes was used to study the electrochemical characteristics of the coating. The results showed that the addition of a certain amount of graphene into the mica/epoxy coating significantly improved the corrosion resistance of the composite coating, and when the graphene content increased, the corrosion resistance of the mica/epoxy coating first increased and then decreased when the corrosion current density of a 35 wt.% 800^#^ mica/epoxy coating with a 0.7 wt.% graphene content was the lowest (7.11 × 10^−1^^3^ A·cm^−2^), the corrosion potential was the highest (292 mV), the polarization resistance was the largest (3.463 × 10^9^ Ω·cm^2^), and the corrosion resistance was improved by 89.3% compared to the coating without graphene. Furthermore, the adhesion of the coating with 0.7 wt.% graphene was also the largest (8.81 MPa, increased by 3.4%) and had the smallest diffusion coefficient (1.566 × 10^7^ cm^2^·s^−1^, decreased by 76.1%), and the thermal stability improved by 18.6%. Finally, the corrosion resistance mechanism of the composite coating with different graphene contents at different soaking times was revealed based on the electrochemistry and morphology characteristics other than water absorption and contact angle.

## 1. Introduction

In the process of oil and gas exploitation, gathering and transportation, there are often many corrosive species in the produced fluid, such as H_2_S, Cl^−^, CO_2_, etc. Metal pipelines suffer from the most common and serious corrosion in corrosive environments containing CO_2_. The leakage of oil and gas caused by the corrosion perforation of the metal pipes not only leads to the waste of resources and the pollution of the environment, but it also causes serious harm to personal safety [1]. The metal pipes are mainly corroded by the charge transfer reaction, which leads to the deterioration of metal surface [2]. Therefore, relevant measures should be adopted to prevent or delay corrosion, such as electrochemical protection, surface treatment, corrosion inhibitor, and coating protection, etc. [3]. Among these measures, coating protection technology is one of the simplest and most effective means, and coatings with epoxy resin as a carrier are widely used.

Epoxy resin (EP) is a thermosetting resin with the excellent comprehensive properties. EP has the characteristics of good adhesion, strong adhesion, outstanding corrosion resistance, excellent chemical resistance, and low shrinkage, and it is easy manufacture and of low cost. Therefore, EP is widely used in coatings, adhesives, and in other fields [4,5,6,7,8]. However, it has many shortcomings, such as its high brittleness, low impact strength, and fatigue resistance, which limit the further application of EP. While many fillers (organic, inorganic, etc.) are introduced into epoxy resin, which can enhance the protection performance of the coating and can further hinder the contact between the metal matrix and the corrosive medium. This has improved the corrosion resistance of pipes and has attracted widespread attention.

Nanomaterials have attracted the attention of many researchers because of their outstanding and unique properties [9], especially graphene, a new material, not only has special lubricating properties, but it has also demonstrated good performance in blocking water molecules, which effectively blocks the diffusion path of corrosive media such as water molecules and oxygen into the coating [10,11,12,13,14]. Graphene can also be added to resin coatings as a nano−filler to improve the mechanical properties and corrosion resistance of the coating [7,15,16]. Cui et al. [17] dispersed graphene (GO) modified by dopamine (DA) into the ethanol and mixed it with epoxy resin. GO-PDA nano–sheets were well dispersed in EP, and the prepared GO-PDA/EP composite coating significantly improved the barrier performance of the waterborne EP coating. Wang et al. [18] found that when the amount of graphene was 2.0 wt.%, the tensile strength and Young’s modulus of the graphene/WPU composite coating increased by 71% and 86%, respectively. Pourhashem et al. [6] developed a new type of nano–solvent epoxy coating with graphene oxide by adding 0.1% graphene oxide to a polymer with a low viscosity and found that the coating had good adhesion, barrier properties, and excellent corrosion resistance. Ye et al. [16] found that graphene oxide (POSS-GO) anticorrosive coatings that had been modified by doping with super–hydrophobic polyhedron oligomeric polysiloxane had good dispersibility and super hydrophobicity, and their synergistic effect enhanced the anticorrosion performance of the composite coating.

Therefore, as a solid lubricating additive, graphene has shown good self-lubricating performance and has also become an ideal anti-corrosion material or filler. In recent years, researchers have found that the addition of graphene into epoxy resin effectively improves the shielding performance, strong acid and alkali resistance, and adhesion of coatings [11,19]. At the same time, it is not easy for graphene to precipitate in organic coatings due its low density [20]. At present, when mica is added to epoxy resin as a filler, the corrosion resistance of its coating is poor, resulting in it being unable to effectively reduce the corrosive medium from infiltrating into the coating. At the same time, the coating will have a stress concentration during curing, resulting in internal defects [9,21,22,23]. When graphene is added to epoxy resin as a filler to prepare an anti-corrosion coating, the coating is prone to agglomerate, increasing internal coating defects. Meanwhile, there are a lot of reports on graphene as a single functional filler being added to epoxy resin, polyurethane, and other polymer resins to improve the wear resistance and corrosion resistance of polymer materials [6,24,25,26], but there are fewer reports on the use of graphene in combination with other fillers in epoxy resins.

In this paper, the mica and graphite power were used as fillers to prepare mica–graphene/epoxy composite coatings with different graphene contents. The effects of different graphene contents on the corrosion resistance of the composite coatings were investigated by electrochemical techniques and high–temperature and high–pressure tests. Combined with the water absorption characteristics and contact angle of the mica–graphene/epoxy, the corrosion resistance mechanism of the composite coatings with different graphene contents at different soaking times was revealed.

## 2. Experimental

### 2.1. Sample Preparation

In this experiment, epoxy resin (E-08) (Shanghai Resin Factory. Shanghai, China) was used as a base material; mica (Chuzhou Gerui Mining Co, LTD. Chuzhou, China) and graphite power (Deyang Carbonene Science &Technology LTD. Deyang, China) were used as fillers; polyether amine (D230) (Wujiang City Blima Technology Industrial Co, Ltd. Wujiang, China) was used as the curing agent; and various additives were added to the mixture to prepare the coating (Nanjing Energy Chemical. Nanjing, China). N80 carbon steel was used as the base metal material in the experiment and had a chemical composition (wt.%) of C: 0.34~0.38, Si: 0.20~0.35, Mn: 1.45~1.75, P: ≤0.15, S: ≤0.15, and Fe balance. The size of the test sample was 100 mm × 50 mm × 3 mm, and the size of the electrochemical test sample was ø20 mm × 3 mm. Among them, the sample used in the electrochemical tests was welded with copper wire and encapsulated with resin, and the reserved working area was 3.14 cm^2^.

The preparation processes to develop the coating in this experiment were divided into two steps: The first step was to prepare a 35 wt.% 800^#^ (800 mesh) mica/epoxy coating, and the specific method was as follows: 35 wt.% 800^#^ mica powder filler was mixed with epoxy resin (E-08), and stirred at 1200 r/min for 1 h, and it was then dispersed by ultrasonication for 1 h so that the filler was fully dispersed in the coating. It was then vacuumized for 30 min, and the mother liquor was prepared. The mother liquor was mixed with polyether amine (D230) in a ratio of 1:0.4, stirred again at 500 r/min for 3 min to ensure that the mixture was uniform, and it was then vacuumized again for 1 min. The second step was to prepare the mica–graphene/epoxy composite coating, and the specific method was as follows: different graphene (graphite powder, natural briquette grade, ~100 nm) contents (0.2, 0.5, 0.7, and 1.0 wt.%) were added to 35 wt.% 800^#^ mica/epoxy coating, stirred at a rotational speed of 1200 r/min for 1 h, dispersed by ultrasonication for 30 min., and then vacuumized for 15 min. Each coating contained a different graphene content and were labeled as TWG-0.2, TWG-0.5, TWG-0.7, and TWG-1.0, respectively. The thickness of the coatings was controlled to 200 ± 10 µm.

### 2.2. Electrochemical Analysis

The corrosion resistance behavior of the mica–graphene/epoxy composite coatings with different graphene contents was studied using a Princeton Electrochemical Workstation (P4000) with a three-electrode system, the auxiliary electrode was a platinum electrode, the reference electrode was a saturated calomel electrode (SCE), and the working electrode was the coating sample. The experiment was carried out under normal pressure. The working electrode was first soaked in 10 wt.% NaCl solution at 80 °C with a total pressure of 5 MPa (CO_2_ partial pressure of 3 MPa) and soaking times of 60, 120, and 240 h, respectively. Before the test, N_2_ was introduced for 2 h to remove oxygen. After the testing system was stabilized in the open circuit potential test in the 10 wt.% NaCl solution, the AC impedance spectrum test was carried out with a sine wave disturbance signal value of 20 mV and a frequency test range of 10^5^~10^−2^ Hz. The test results were fitted using the ZSimpWin software, ensuring that the overall coincidence value was less than 0.01, and the relative fitting error value of each circuit element in the equivalent circuit was less than 10%. During the polarization curve test, the potential scanning interval was −0.5~0.5 V, and the scanning rate was 0.5 mV·s^−1^. The polarization curve was fitted with the workstation software.

### 2.3. Simulating Immersion Test

A high-temperature and high-pressure autoclave was used to simulate the underground operation environment, and the experimental temperature was 120 °C, the total pressure was 20 MPa, and the CO_2_ partial pressure was 5 MPa. The solution was 10 wt.% NaCl. The test parameters were set as shown in Table 1.

### 2.4. Adhesion Test

The adhesion was tested according to the standard GB/T 5210-2006 [27]. Each group of samples was tested three times on average to ensure the accuracy of the experimental data.

### 2.5. Water Absorption Rate

An electronic balance with an accuracy of (0.0001 g) was used for the weighing procedures, and the test method was as follows: the samples before immersion were weighed and recorded as the initial weight (m_0_). Then, the samples were soaked in a high-temperature autoclave with the setting parameters of 80 °C and total pressure of 5 MPa (the partial pressure of CO_2_ was 3 MPa) for different times, and they were then taken out, and filter paper was used to dry any of the remaining solution on the surface of the sample. The weight was taken and was denoted as m_t_. Three parallel samples were measured in each group, and the mean value was taken as the final weight. The water absorption rate was calculated according to Equation (1):(1)$$Q_{t}=\frac{m_{t}-m_{0}}{m_{0}}\times 100\%$$
where: Q_t_ represents the water absorption rate at time t; m_t_ represents the weight of the sample at time t; and m_0_ represents the initial weight of the sample.

### 2.6. Contact Angle

A JC2000D1 contact angle tester was used to test the contact angle of the coating in the solution; the needle was 0.725 mm and the droplet volume was 10 mL. Each sample was tested three times, and the average contact angle of the left and right droplets was taken as the contact angle of the coating.

## 3. Results and Discussion

### 3.1. Electrochemical Impedance

Electrochemical impedance is the most commonly used method for the detection and evaluation of coating failure and shielding performance. The electrochemical information of the fillers and metal matrix can be obtained by fitting and analyzing the coating with an equivalent circuit diagram [28]. Figure 1 shows the Nyqusit curves of the composite coatings with different graphene contents at different immersion times, and its equivalent circuit diagram is shown in Figure 1d, in which *R*_s_ represents the solution resistance, *C*_c_ and *R*_c_ represent the coating capacitance and coating resistance, and *C*_dl_ and *R*_ct_ represent the double layer capacitance and charge transfer resistance [29], and the fitting data results are shown in Table 2. It can be seen from Figure 1 and Table 2 that the composite coatings showed double capacitive reactance characteristics after immersion for different times. After immersion for 60 h, the capacitive arc radius of the composite coating first increased and then decreased as the amount of graphene increased, as shown in Figure 1a, where the resistance *R*_ct_ of the epoxy varnish was 1.620 × 10^9^ Ω·cm^2^, and when 0.7 wt.% graphene was added, the resistance *R*_ct_ of the composite coating reached the maximum of 5.65 × 10^10^ Ω·cm^2^. When the amount of graphene increased to 1.0 wt.%, the resistance *R*_ct_ of the composite coating began to decrease to 3.167 × 10^9^ Ω·cm^2^, and the order of the composite coating resistance was TWG-0.7 > TWG-0.5 > TWG-0.2 > TWG > TWG-1. Figure 1b,c shows the Nyqusit curves of the composite coating after immersion for 120 and 240 h, which show the same laws as those after immersion for 60 h. The impedance at low frequencies is another parameter that is commonly used to evaluate corrosion protection; evidently, the corrosion resistance of the TWG-0.7 sample was the best. Yang [30] thought that 0.7 wt.% amino-functionalized graphene oxide (AGO) exhibited the best anti-corrosion properties, the *R*_ct_ of which was only 3.80 × 10^6^ Ω·cm^2^. However, the *R*_ct_ of the OG/EP coating remained at 10^11^ Ω cm^2^ after 7 days of immersion in 3 MPa pure O_2_ and in a 3.5 wt.% NaCl coupling environment [31].

The radius of the capacitance arc represents the strength or weakness of the corrosion resistance of the coating to the corrosive medium [32,33], while the resistance *R*_ct_ reflects the difficulty of the charge transfer on the metal interface [34], which can be used to evaluate the corrosion rate of the metal matrix. Generally speaking, the higher the *R*_ct_ is, the more difficult the corrosion reaction is, and the slower the corrosion rate is. On the contrary, the lower the *R*_ct_ is, the more likely a corrosion reaction is, and the faster the corrosion rate is [35]. In addition, *R*_c_ is the coating resistance, which reflects the quality of the coating. The lower the porosity per unit area of the coating is, the higher the *R*_c_ is, and the better the corrosion resistance is.

When the graphene content increased (0.2 wt.%~0.7 wt.%), the values of *R*_c_ and *R*_ct_ increased in different degrees after immersion for 60, 120, and 240 h, and were higher than those without graphene. However, when the addition amount of graphene was 1.0 wt.%, the *R*_c_ and *R*_ct_ values of the coating decreased, but they were always higher than those of the coating without graphene, which can be attributed to the fact that the lamellar structure of graphene is stacked layer by layer in the coating, meaning that the dense physical protective layer that is formed can isolate the contact between the matrix and the outside corrosive medium well, and the uniform distribution of the unique lamellar structure of graphene reduces the internal porosity of the composite coating [36,37,38,39], which slows down the diffusion rate of the electrolyte molecules in the coating and makes it be difficult for the corrosive medium to penetrate into the coating, demonstrating an anticorrosion effect on the coating. Therefore, adding nano-fillers to epoxy coatings can effectively enhance the anticorrosion performance [40]. However, when the amount of graphene exceeded a certain amount, the uneven graphene dispersion in the coating caused the agglomeration, leading to an increase in the defective graphene aggregates, cracks, holes, and other defects, and the barrier effect on the corrosive media was weakened [41].

### 3.2. Polarization Curve

Figure 2 shows the polarization curves of the composite coatings measured at different immersion times. The polarization curves were fitted in the Tafel area using the ZSimpWin software, and the self-corrosion potential *E*_corr_ and self-corrosion current density *i*_corr_ are shown in Table 3.

It can be seen from Figure 2 and Table 3 that with the increasing graphene content, both the self-corrosion potential *E*_corr_ and polarization resistance *R*_p_ of the composite coating first increased and then decreased, while the self-corrosion current density *i*_corr_ showed an opposite trend. For example, after adding graphene (0.2~0.7 wt.%), the self-corrosion potential *E*_corr_ of the composite coating increased by about 20 mV compared to that of the coating without graphene, indicating that the appropriate amount of graphene protected the matrix of the composite coating to some extent. Kumar [24] et al. also obtained the same result when the proper amount of graphene resulted in the self-corrosion potential of the composite coating obviously increasing, that is to say, the corrosion resistance of the composite coating was enhanced due to the addition of the proper amount of graphene, which was determined based on the corrosion thermodynamics.

At the same immersion time, for example, after immersion for 60 h, the self-corrosion potential of the coating without graphene was 46 mV, and the self-corrosion current density was 2.19 × 10^−1^^1^ A·cm^−2^. As the graphene content increased, the self-corrosion potential of the coating increased, and the self-corrosion current density decreased. When the graphene content in the coating was 0.7 wt.%, the self-corrosion potential of the composite coating was 292 mV, which was higher than that (−332 mV) presented in reference [30], indicating the corrosion tendency decreased from the perspective of corrosion thermodynamics. Additionally, the self-corrosion current density was 7.11 × 10^−1^^3^ A·cm^−2^, which was lower than that (1.15 × 10^−7^ A·cm^−2^) presented in Reference [30], indicating that the corrosion rate decreased from the perspective of corrosion kinetics. When the graphene content increased to 1.0 wt.%, the self-corrosion potential of the coating was only 76 mV, which was lower than that of 0.7 wt.%, and the self-corrosion current density was 1.65 × 10^−1^^1^ A·cm^−2^. The results after immersion for 120 and 240 h were basically consistent with those after immersion for 60 h. Generally, the lower the corrosion potential is, the smaller the corrosion current density is, and the better the corrosion resistance of the matrix is [42,43,44].

At different immersion times, the coating with 0.7 wt.% graphene (TWG-0.7) had the lowest self-corrosion current density, the highest corrosion potential, and the largest polarization resistance, indicating that, when 0.7 wt.% graphene was added to the 35 wt.% 800^#^ mica–graphene/epoxy coating, its corrosion resistance was the best. However, when excessive graphene was added (1.0 wt.%) to the coating, the internal defects of the coating might have increased, resulting in a decrease in the anticorrosion performance of the coating. Therefore, the composite coating with 0.7 wt.% graphene obtained the best corrosion resistance, which is consistent with the results obtained by electrochemical impedance.

### 3.3. Morphology Characteristics

The corrosion resistance of the composite coating was further analyzed by high-temperature autoclave, and the test parameters were set as shown in Table 1.

Figure 3 shows the macroscopic morphology of the composite coating before and after immersion at high temperatures and high pressure for 240 h, and it can be seen from Figure 3A that the surface of the composite coating before immersion was flat and defect-free, its color was black and bright, and its luster was full. After immersion for 240 h, the gloss of the composite coating decreased, and chromatic aberration was present. Furthermore, bubbles appeared on the surface of the composite coating without graphene (TWG) and with different graphene contents (TWG-0.2, TWG-0.5, and TWG-1.0), and the order of the size and number of the bubble was TWG-0.5 > TWG-0.2 > TWG > TWG-1.0, while there were no bubbles or cracks on the surface of the coating with 0.7 wt.% graphene (TWG-0.7).

Graphene can effectively fill defects such as pores and cracks in the epoxy coating due to its small size [37], reduce the diffusion rate of the corrosive medium in the coating, and enhance the impermeability of the coating. Therefore, the addition of a proper amount of graphene improved the corrosion resistance of the coating, and the corrosion resistance of the coating with 0.7 wt.% graphene was the best. Yang [30] proved that the corrosion resistance increased due to the graphene barrier. However, this was not conducive to the addition of the excessive graphene to achieve better dispersion and filling in epoxy resins, which reduced the impermeability and corrosion resistance of the coating.

### 3.4. Water Absorption and Hydrophobicity

Figure 4 shows the water absorption curves of the composite coatings with different graphene contents after immersion for 240 h at 80 °C and 5 MPa (3 MPa CO_2_ partial pressure). The test solution was 10% NaCl, and the sampling period was 24 h. It can be seen that the order of the final saturated water absorption of the coating was TWG-0.7 (1.153%) < TWG-0.5 (1.427%) < TWG-0.2 (1.576%) < TWG (2.03%) < TWG-1 (2.757%).

When the amount of graphene was 0.2~0.7 wt.%, the water permeability of the composite coating was higher than that of the 35 wt.% 800^#^ mica/epoxy coating, and when the amount of graphene was 0.7 wt.%, the water absorption of the composite coating was the smallest, and the water permeability of the composite coating was stronger against the corrosive medium. However, when the amount of graphene was 1.0 wt.%, the water absorption of the coating increased, further indicating that the ways through which the corrosive medium could be diffused into the coating increased.

In order to investigate the influence of different amounts of graphene on the hydrophobic properties of the composite coating, the wetting contact angles of the mica–graphene/epoxy composite coatings before and after immersion were tested, as shown in Table 4.

It can be seen from Table 4 that before immersion, the contact angle of the composite coating without graphene was 72.8° ± 0.16°, and the contact angle increased as the graphene content in the composite coating increased. After immersion for 240 h, the contact angle of the composite coating first increased and then decreased as the amount of the graphene in the composite coating increased, and the maximum contact angle (88.2° ± 0.08°) was obtained when the graphene content was 0.7 wt.%. According to the pre-evaluation index of the interface contact angle, the larger the contact angle of the filler is, the worse the hydrophilicity is, while the better the lipophilicity and hydrophobicity are, the easier it is for it to disperse in organic matter, leading to good compatibility with organic matter, and vice versa. It can be seen that the composite coating with 0.7 wt.% graphene had the best hydrophobic properties after immersion, the spatial structure between graphene and mica was reasonable, and the corrosive medium was better isolated. At the same time, the hydrophobic interactions were partially contributed to by π–π electron donor–acceptor interactions between the graphene surface of the activated carbon (AC) and the corrosive spices [23]. Additionally, then, the corrosion resistance of the coating was significantly improved. Therefore, the impermeability of graphene plays a critical role when employing it in corrosion barrier coatings to effectively blocking reactive gases, fluids, salts, and acids [45]. The water absorption kinetics curve of the mica–graphene/epoxy composite coating was fitted by the corresponding curve, as shown in Figure 5, and the results are shown in Table 5. It can be seen that the diffusion coefficient of the mica–graphene/epoxy composite coating was 5.306 × 10^7^ cm^2^·s^−1^. As the graphene content increased, the diffusion coefficient of the mica–graphene/epoxy composite coating first decreased and then increased. When 0.7 wt.% graphene was added, the diffusion coefficient of the coating reached its lowest value of 1.566 × 10^7^ cm^2^·s^−1^. With the further addition of graphene (1.0 wt.%), the diffusion coefficient of the coating increased to 7.526 × 10^7^ cm^2^·s^−1^. Furthermore, it was found that the diffusion coefficient of the composite coating to the aqueous solution was smaller when the amount of graphene was in the range of 0.2~0.7 wt.%, but it was larger when the amount of graphene was 1.0 wt.%. The smaller the diffusion coefficient is, the better the impermeability of the coating is. Therefore, the addition of the proper amount of graphene can effectively fill defects such as pores and cracks in the coating, enhance the impermeability of the coating, and delay the contact between the corrosive medium and the metal matrix, enhancing the corrosion resistance of the coating. The composite coating with 0.7 wt.% graphene had the best impermeability. However, after adding excessive graphene, the graphene was easy to agglomerate, and the internal defects of the composite coating increased, which provided favorable conditions for the penetration of the corrosive medium and the corrosion of metal matrix and reduced the impermeability of the composite coating. Finally, the corrosion resistance of the coating was reduced. At this time, the impermeability of mica–graphene/epoxy coating showed a negative growth trend, and it was easy for the corrosive medium to penetrate into the coating and react with the matrix, thus accelerating the corrosion of the metal matrix.

### 3.5. Adhesion

The change rule of the adhesion of the composite coating with different graphene contents after immersion for different times was explored; Table 6 shows the adhesion changes that took place in the composite coatings before and after immersion at 80 °C for 60 and 120 h. It can be seen that the change rule of the adhesion of the composite coating was TWG-0.7 > TWG-0.5 > TWG-0.2 > TWG > TWG-1.0.

It can be seen from Table 6 that before immersion, the adhesion of the coating first increased and then decreased as the graphene content increased. The addition of the proper amount of graphene was beneficial to improving the adhesion of the composite coating. For example, before immersion, the adhesion of the coating both with and without the different graphene contents (0.2~0.7 wt.%) was 8.64, 8.71, 8.79, and 8.81 MPa. The law of the coating after immersion for 60 and 120 h was the same as that before immersion. Therefore, the adhesion was the highest when the graphene content in the coating was 0.7 wt.%. Additionally, detaching status images of the TWG-0.7 samples were shown in Figure 6. It can be seen that the detaching areas of the composite coating before immersion was tiny. Evidently, the composite role of mica and graphene has the capability to improve the adhesion strength to the matrix. Shaker [46] thought that epoxy/graphene composite coatings were well attached to the substrate and that there were not any gaps. Additionally, the detaching areas increased as the immersion time increased, which further sustained the adhesion results mentioned above. Cai [21] argued that the good interface compatibility between the filler (such as modified mica) and the matrix meant that the coating had high anticorrosion performance.

The graphene particle size (graphite power: ~100 nm) was small, so the particles could fill the pores in mica and reduce the internal pore defects of the composite coating. Graphene was flexibly wound with the epoxy polymer chains in the composite coating, which increased the contact area between the composite coating and the substrate surface [47]. Additionally, the oxygen-containing functional groups in the graphene could easily form covalent bonds with the matrix [48], making the adhesion of the composite coating increase. However, when too much graphene was added (more than 0.7 wt.%), the graphene resulted in the number of divided mica units increasing, increased the internal defects in the composite coating, decreased the intermolecular force between the composite coating and the substrate force [49], and then decreased the adhesion of the coating with the matrix.

In addition, it can be seen from Table 6 that the adhesion of the coating without graphene was 8.64 MPa before immersion, and as the immersion time increased, the adhesion of the composite coating gradually decreased from 7.42 to 7.26 MPa. For composite coatings with a certain amount of graphene, the results were consistent with those of the coating without graphene. Therefore, as the immersion time decreased, the more the corrosive medium permeated into the coating, more ion transport channels were formed, resulting in an increase in the internal defects of the composite coating, a decrease in shielding against the corrosive medium, and a decrease in the adhesion between the coating and the metal substrate [48,50,51].

## 4. Corrosion Resistance Mechanism

The adhesion of a coating treated with a corrosive medium can reflect the anti-corrosion performance of the coating because the corrosive medium will react with the metal matrix after passing through the coating, thus weakening the adhesion of the coating with the metal matrix. It can be seen from Table 6 that the coating with 0.7 wt.% graphene had the highest adhesion; therefore, its protective performance was the best.

Based on the analysis mentioned above, the composite coating prepared by adding the appropriate amount of graphene into the mica/epoxy coating had a certain degree of hydrophobicity, and the surface of the coating had a small contact area with the corrosive medium, which reduced the infiltration of the corrosive medium into the coating to a certain extent. At the same time, a small amount of graphene was able to disperse itself well within the coating, and graphene was evenly distributed among the mica sheets, keeping the layers tightly stacked. At the same time, the advantage of the small size of the graphene was that the particles could fill in the internal porosity and defects of the coating, which made it more difficult for the corrosion medium to penetrate into the coating, resulting in the formation of a more complex anti−corrosion path.

According to the results of the high temperature and high pressure immersion test, as shown in Figure 7a, when the amount of graphene was less than 0.7 wt.%, there were tiny bubbles on the contact surface between the corrosive medium and the coating, the reason for which is that epoxy resin is sensitive to the temperature and pressure range, so the internal pores of the coating cannot be completely filled by only adding a small amount of graphene, and graphene has good conductivity. After the addition of a small amount of graphene into the coating, the amounts of the epoxy resin and curing agent in the composite coating were still higher. The cross–linking reaction took place easily, forming a coating with good compactness, while the chemical reaction produces oxygen and other substances, which fill in pores and are not easy to diffuse out, resulting in bubbling in the coating, but no graphene agglomeration occurs in the coating.

When the amount of graphene was 0.7 wt.%, there was no bubbling at the contact interface between the coating and the corrosive medium, as shown in Figure 7b, and the graphene was evenly distributed in the coating, resulting in good corrosion resistance. Figure 7c shows that when the graphene content was more than 0.7 wt.%, there was obvious bubble formation. However, it was not easily wetted or sufficiently dispersed for the excessive graphene due to its huge specific surface area [22], meaning that the graphene would be randomly distributed in the coating due to the graphene agglomerates decreasing the energy of the system. The increase in the orientation degree between the graphene sheets in the composite coatings and the substrate quickened the diffusion process and flux [46]. Furthermore, the conductivity between the matrix and the corrosive medium increased, thus forming a good primary cell effect. Graphene can quickly transfer electrons and accelerate corrosion reactions between the metal matrix and the corrosive medium. Gases, such as H_2_, which are generated by electrochemical reaction cannot diffuse through corrosion channels in time, thus resulting in gathering and bubbling in the composite coating. The water absorption performance of the coating was enhanced, and the coating was obviously loose when it was in contact with the metal matrix and accompanied by the generation of corrosion products, and the adhesion decreased. It is easy to accelerate the occurrence of corrosion for long-term immersion in a corrosion solution. According to the results of the electrochemical EIS test, the resistance of the mica–graphene/epoxy coating decreased from 1.872 × 10^10^ Ω·cm^2^ to 4.333 × 10^9^ Ω·cm^2^ after immersion for 60 h; the resistance of the composite coating decreased from 1.537 × 10^9^ to 4.621 × 10^8^ Ω·cm^2^ after immersion for 120 h; and the resistance of the composite coating decreased from 4.130 × 10^8^ to 1.321 × 10^8^ Ω·cm^2^ after immersion for 240 h, but they were much higher than 10^6^ Ω·cm^2^.

In the same immersion time, as the graphene content increased, the resistance of the coating first increased and then decreased. It can be seen that the addition of an appropriate amount of graphene into a mica/epoxy coating can not only play an excellent physical isolation role, but it can also increase the compactness of the coating, resulting in significant improvements in the anti-permeability performance of the composite coating, in prolonging the contact time between the corrosive medium and the metal substrate, and in achieving the effect of improving the corrosion resistance of the coating.

From the test results obtained from the polarization curves, it can be also seen that with the extension of the immersion time, the self−corrosion potential of the coating decreased correspondingly, the corrosion current density increased, the polarization resistance decreased, and the corrosion resistance of the material decreased. However, as the graphene content in the composite coating increased at the same immersion time, the corrosion potential increased, the polarization resistance increased, the corrosion current density decreased, and the corrosion resistance of the coating increased.

Therefore, when graphene and mica were incorporated into composite coatings, a chemical interaction took place between nano–silica (Mica) and graphene [52], and the intensity of the C–O stretch in the COOH group and carboxylic functional group was strengthened by the addition of graphene sheets to the epoxy matrix [46]. Besides serving as a labyrinth physical barrier [53], graphene also reduced the pore defects in the organic coatings [15], and the protective properties of the composite coating on the matrix were enhanced. As a result, corrosion is estimated to initiate after longer periods of time, allowing the underlayer to enjoy a prolonged service life.

## 5. Conclusions

(1)The addition of graphene effectively improved the anti-corrosive properties of the composite coatings based on corrosion thermodynamics and kinetics.(2)The addition of 0.7 wt.% graphene maximizes the impermeability of graphene to inhibit the diffusion of corrosive media and also result in good surface hydrophobicity.(3)The addition of the proper amount of graphene to the mica–epoxy coating served as a labyrinth physical barrier along with mica and reduced the pore defects in the organic coatings by reacting with epoxy and other filler, prolonging the corrosion initiation time and the development of the metallic matrix.

## Figures and Tables

**Figure 1 materials-15-01194-f001:**
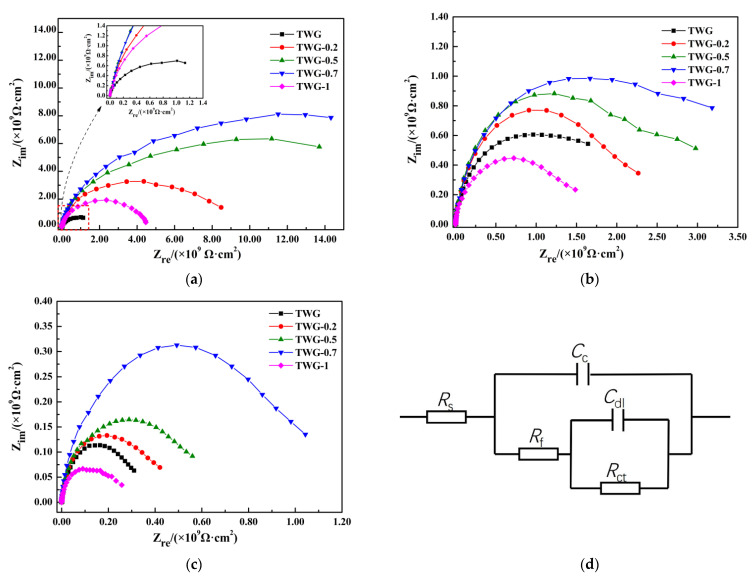
Nyquist diagram and EIS equivalent circuit diagram of composite coating for different immersion times. (**a**) After immersion for 60 h. (**b**) After immersion for 120 h. (**c**) After immersion for 240 h. (**d**) EIS equivalent circuit diagram.

**Figure 2 materials-15-01194-f002:**
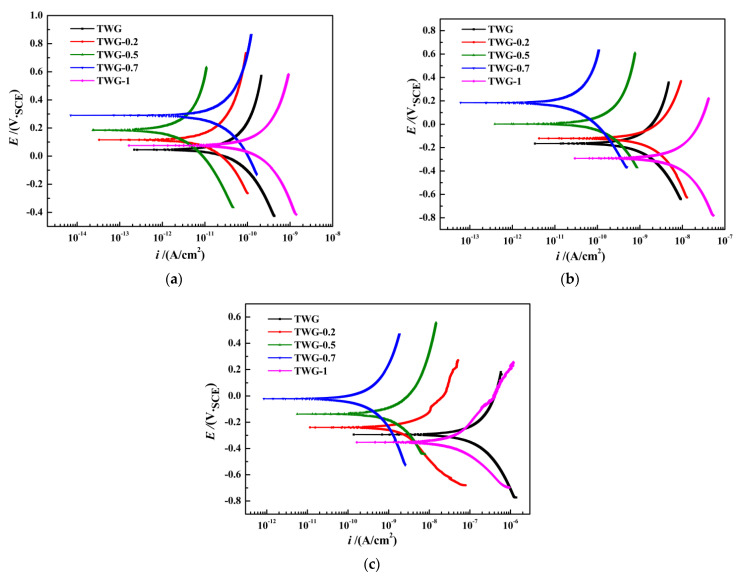
Polarization curves of the composite coating after immersion for different times. (**a**) After immersion for 60 h. (**b**) After immersion for 120 h. (**c**) After immersion for 240 h.

**Figure 3 materials-15-01194-f003:**
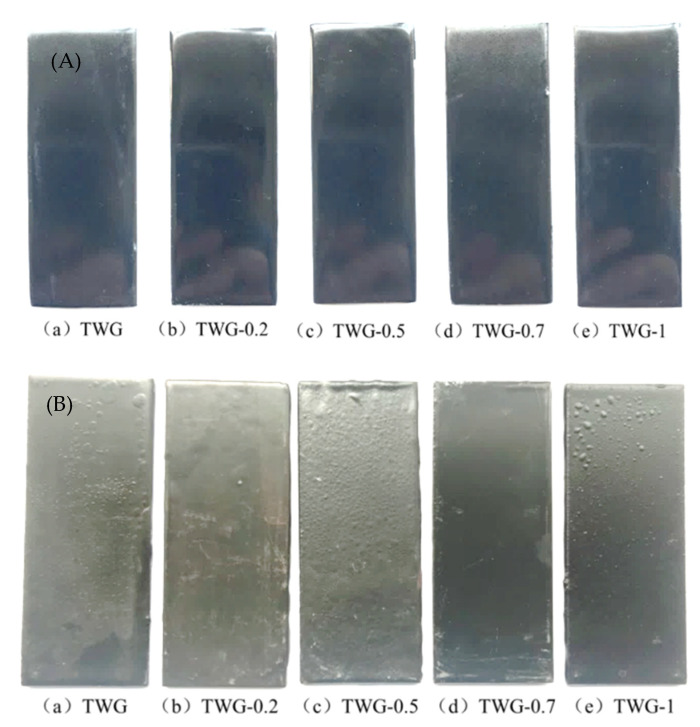
Macro–morphology of composite coating before and after immersion at high temperatures and high-pressure levels for 240 h: (**A**) before immersion and (**B**) after immersion.

**Figure 4 materials-15-01194-f004:**
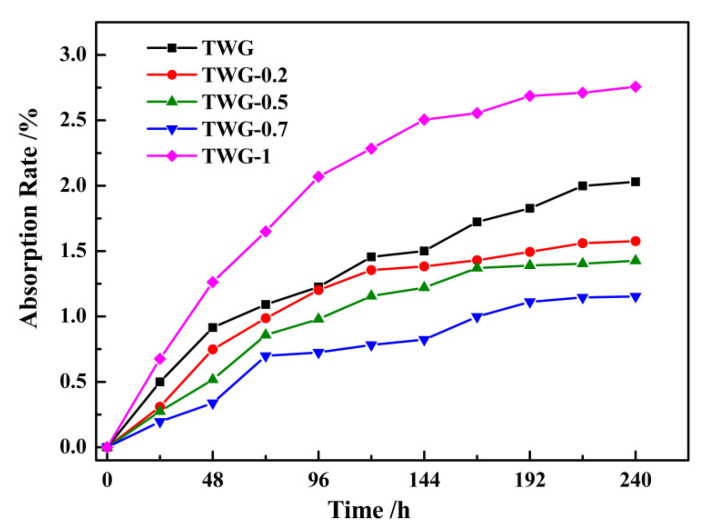
Water absorption curve of mica/graphene–epoxy composite coating.

**Figure 5 materials-15-01194-f005:**
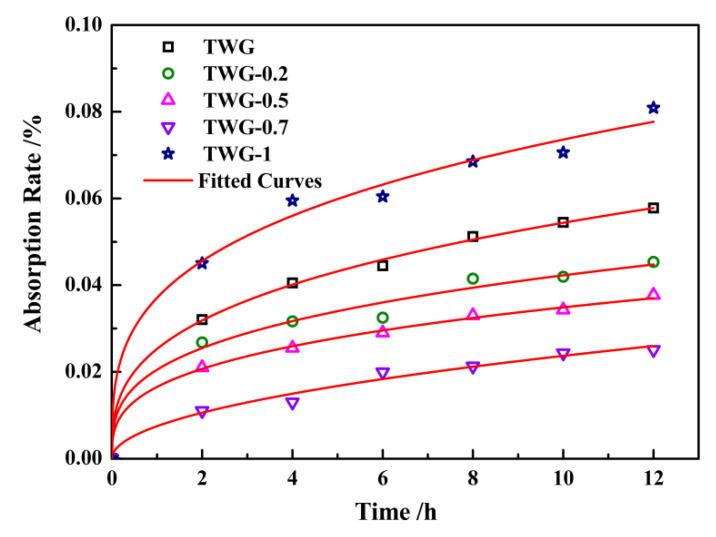
Water absorption kinetics curve of the coating at the initial soaking stage.

**Figure 6 materials-15-01194-f006:**
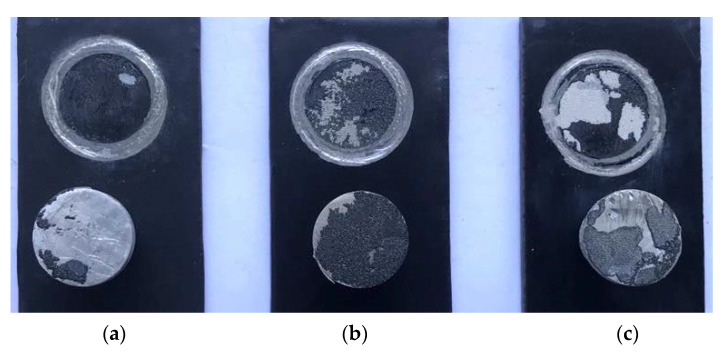
Detaching status macro–images of the composite coatings with 0.7 wt.% graphene before and after immersion at 80 °C for 60 and 120 h. (**a**) 0 h. (**b**) 60 h. (**c**) 120 h.

**Figure 7 materials-15-01194-f007:**
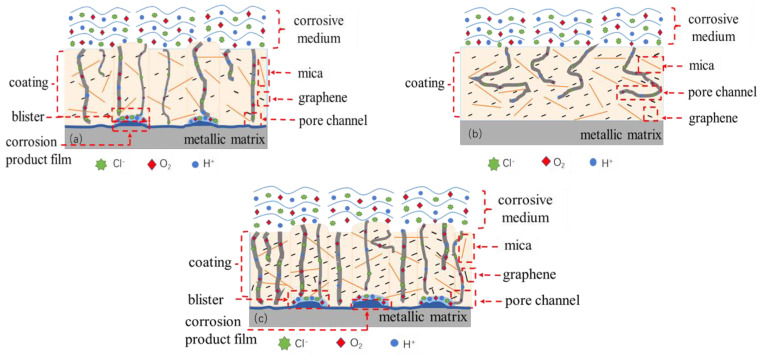
Schematic diagram of mica–graphene/epoxy composite coating with different addition amount of graphene for isolating corrosion medium infiltration. (**a**) Less than 0.7%. (**b**) 0.7% graphene. (**c**) More than 0.7 wt.%.

**Table 1 materials-15-01194-t001:** High-temperature and high-pressure corrosion test parameters.

Temperature	Test Solution	Total Pressure	Test Time
144 ℃	10 wt.% NaCl	20 MPa	240 h

**Table 2 materials-15-01194-t002:** Fitting data of the equivalent circuit diagrams.

Immersion Time	Sample	*R*_s_/(Ω·cm^2^)	*C*_c_/(F·cm^−2^)	*R*_c_/(Ω·cm^2^)	*C*_dl_/(F·cm^−2^)	*R*_ct_/(Ω·cm^2^)
60 h	TWG	1.220 × 10^−3^	1.076 × 10^−10^	1.212 × 10^9^	9.984 × 10^−10^	1.620 × 10^9^
	TWG-0.2	2.295 × 10^−2^	9.716 × 10^−11^	7.477 × 10^9^	1.139 × 10^−11^	2.539 × 10^9^
	TWG-0.5	1.167 × 10^−2^	5.056 × 10^−11^	1.559 × 10^10^	3.851 × 10^−11^	4.167 × 10^9^
	TWG-0.7	1.487 × 10^−2^	2.421 × 10^−11^	1.872 × 10^10^	4.132 × 10^−11^	5.65 × 10^9^
	TWG-1	8.302 × 10^−2^	9.958 × 10^−11^	4.333 × 10^9^	3.641 × 10^−11^	3.167 × 10^9^
120 h	TWG	1.997 × 10^−3^	2.323 × 10^−10^	9.164 × 10^8^	1.024 × 10^−9^	1.393 × 10^9^
	TWG-0.2	2.575 × 10^−2^	2.062 × 10^−10^	9.646 × 10^8^	1.064 × 10^−9^	1.486 × 10^9^
	TWG-0.5	7.341 × 10^−3^	1.648 × 10^−10^	1.281 × 10^9^	6.867 × 10^−10^	1.810 × 10^9^
	TWG-0.7	6.704 × 10^−2^	9.115 × 10^−11^	1.537 × 10^9^	9.861 × 10^−10^	2.966 × 10^9^
	TWG-1	4.444 × 10^−3^	4.182 × 10^−10^	4.621 × 10^8^	8.419 × 10^−10^	8.73 × 10^8^
240 h	TWG	2.472 × 10^−3^	3.866 × 10^−10^	2.182 × 10^8^	6.814 × 10^−9^	1.129 × 10^8^
	TWG-0.2	1.127 × 10^−2^	3.579 × 10^−10^	2.569 × 10^8^	1.986 × 10^−10^	2.393 × 10^8^
	TWG-0.5	3.682 × 10^−2^	2.619 × 10^−10^	3.137 × 10^8^	4.578 × 10^−10^	6.532 × 10^8^
	TWG-0.7	1.252 × 10^−3^	1.273 × 10^−10^	4.130 × 10^8^	8.912 × 10^−10^	7.404 × 10^8^
	TWG-1	3.357 × 10^−3^	5.047 × 10^−9^	1.321 × 10^8^	2.299 × 10^−10^	4.790 × 10^8^

**Table 3 materials-15-01194-t003:** Polarization curve fitting data of composite coating immersed for different times.

Immersion Time	Sample	*E*_corr_/mV	*i*_corr_/(A·cm^−2^)	*R*_p_/(Ω·cm^2^)
60 h	TWG	46	2.19 × 10^−11^	6.324 × 10^8^
	TWG-0.2	116	3.29 × 10^−12^	1.695 × 10^9^
	TWG-0.5	186	2.36 × 10^−12^	2.589 × 10^9^
	TWG-0.7	292	7.11 × 10^−13^	3.463 × 10^9^
	TWG-1	76	1.65 × 10^−11^	8.171 × 10^8^
120 h	TWG	−165	3.41 × 10^−12^	9.87 × 10^7^
	TWG-0.2	−121	4.26 × 10^−12^	2.239 × 10^8^
	TWG-0.5	18	3.89 × 10^−13^	5.761 × 10^8^
	TWG-0.7	184	6.09 × 10^−14^	7.280 × 10^8^
	TWG-1	−292	2.96 × 10^−11^	4.089 × 10^7^
240 h	TWG	−294	1.38 × 10^−10^	9.587 × 10^6^
	TWG-0.2	−239	1.16 × 10^−11^	1.212 × 10^7^
	TWG-0.5	−138	5.63 × 10^−12^	4.679 × 10^7^
	TWG-0.7	−21	8.53 × 10^−13^	1.747 × 10^8^
	TWG-1	−353	1.65 × 10^−10^	5.292 × 10^6^

**Table 4 materials-15-01194-t004:** Contact angles and pictures of composite coatings before and after immersion with different graphene addition amounts.

Coating	Before Immersion	After Immersion for 240 h
TWG	72.8° ± 0.16°	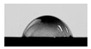	68.3° ± 0.14°	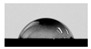
TWG-0.2	90.35° ± 0.07°	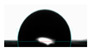	86.1° ± 0.08°	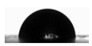
TWG-0.5	91.5° ± 0.16°	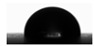	86.5° ± 0.14°	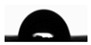
TWG-0.7	92.7° ± 0.08°	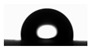	88.2° ± 0.08°	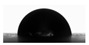
TWG-1.0	93.1° ± 0.16°	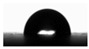	86.4° ± 0.16°	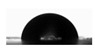

**Table 5 materials-15-01194-t005:** Nonlinear fitting results and diffusion coefficients of the coatings.

Sample	Pressure	A	B	Diffusion Coefficient/cm^2^ s^−1^
TWG	5 MPa	0.02442	0.54675	5.306 × 10^7^
TWG-0.2		0.2245	0.36104	3.443 × 10^7^
TWG-0.5		0.01641	0.48918	2.948 × 10^7^
TWG-0.7		0.00847	0.56439	1.566 × 10^7^
TWG-1.0		0.03996	0.33565	7.526 × 10^7^

**Table 6 materials-15-01194-t006:** Experimental data for composite coating adhesion before and after immersion.

Sample	Before Soaking Adhesion/MPa	After Immersion for 60 h Adhesion/MPa	After Immersion for 120 h Adhesion/MPa
TWG	8.64 ± 0.07	7.42 ± 0.10	7.26 ± 0.08
TWG-0.2	8.71 ± 0.16	7.49 ± 0.15	7.39 ± 0.07
TWG-0.5	8.79 ± 0.08	7.54 ± 0.12	7.43 ± 0.08
TWG-0.7	8.81 ± 0.04	7.69 ± 0.15	7.51 ± 0.09
TWG-1.0	8.32 ± 0.07	5.53 ± 0.08	3.30 ± 0.13

## Data Availability

The data presented in this study are available on request from the corresponding author.

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
