# Peer review of "Corrosion Resistance Mechanism of Mica–Graphene/Epoxy Composite Coating in CO2-Cl− System"

_materials, 2022, doi:10.3390/ma15031194_

Round 1
Reviewer 1 Report
The current article aims to develop and characterize different graphene-mica/epoxy composite coatings proposed as a solution to improve the lifetime, as well as corrosion resistance, of metal pipelines for oil and gas transportation industry. The subject is relevant for science and practice but the manuscript has numerous misspellings. Hence, the current form of the manuscript requires major revisions, extensive editing of English grammar and style is also required.
Some observations are summarized below:
- Please carefully proof-read spell check to eliminate grammatical and spelling errors like: the coating sample, Euqation 1, mica-graphene/graphen, etc.
- In the manuscript it is stated that other various additives were added to prepare the coating. Please mention all the information regarding the manufacturing process of the samples, as well the provider for all the involved materials.
- It is not clear what does 35 wt. % 800# mica/epoxy coating mean? What does the 800# represent? If it refers to the mesh number for sieving the mica particles, please specify. Moreover, the dimension of the graphene particles/fibers it is not stated?
- How were the graphene and mica particles mixed with the epoxy resin?
- In the case of water absorption rate determination, the immersion medium is not stated. I suppose it is the same 10wt.% NaCl solution.
- Corrosion current density is usually denoted using icorr, while the corrosion current is indicated with Icorr.
- Did the authors measure the electrical conductivity of such coatings?
- Why did the authors not investigate the bulk epoxy coating without mica or graphene, as reference material in similar testing conditions?
- In the abstract it is stated that the surface characteristics were observed by scanning electron microscopy. The current manuscript does not provide any SEM micrographs, although some aspects about pores and graphene distribution in the coating are discussed, without being sustained. (e.g. …excessive graphene, which were randomly distributed in the coating).
- Figure 2 (c) has an inferior quality compared to the other 2 related figures (2a and 2b). Furthermore, in figure 2a, the logarithmic axis has different values (ranging from 1014 to 108) compared to the vales reported in Table 2, where there is a minus sign at the exponential factor. Please check those values again.
- Moreover, I find figure 3 redundant, as it has a low quality and it does not offer any significant information.
- The adhesion test should be sustained with some optical/SEM micrographs.
- Additionally, I recommend an article on similar investigations, that can be consulted: Cai, Y.; Meng, F.; Liu, L.;Liu, R.; Cui, Y.; Zheng, H.; Wang, F. The Effect of the Modification of Mica by High-Temperature Mechanochemistry on the Anticorrosion Performance of Epoxy Coatings. Polymers 2021, 13, 378.
Author Response
Reviewer 1:
- Please carefully proof-read spell check to eliminate grammatical and spelling errors like: the coating sample, Euqation 1, mica-graphene/graphen, etc.
Response: Thanks for your review and suggestions, the errors have been revised in the whole manuscript. Please see Page 4 and Page 7. And English style and grammar has been edited by the editing services listed at https://www.mdpi.com/authors/english, Please see the editing certificate.
- In the manuscript it is stated that other various additives were added to prepare the coating. Please mention all the information regarding the manufacturing process of the samples, as well the provider for all the involved materials.
Response: According to your advices, the manufacturing processes of the samples and all the involved materials have been listed in the manuscript. Please see Paragraphs 1 and 2 in Page 3.
- It is not clear what does 35 wt. % 800# mica/epoxy coating mean? What does the 800# represent? If it refers to the mesh number for sieving the mica particles, please specify. Moreover, the dimension of the graphene particles/fibers it is not stated?
Response: 35wt.% in 35 wt. % 800# mica/epoxy represents the mica content added into the mica/epoxy coating, and 800# represents the mesh of mica particles. Additionally, the dimension of the graphene particles (power) was smaller than 100 nm (lamellar thickness: < 1.5nm, specific surface area: 300~400 m2/g), which has been added in the manuscript. Please see Paragraph 2 in Page 3.
- How were the graphene and mica particles mixed with the epoxy resin?
Response: The preparation process of the coating in this experiment was divided into two steps. The first step was to prepare 35 wt.% 800 mesh (800#) mica/epoxy coating, and the specific method was as follows: 35 wt.% 800# mica powder filler was mixed with EP (E-08), stirred at 1200 r/min for 1 h, and dispersed by ultrasonic for 1 h, such that the filler was fully dispersed in the coating; in sequence, it was vacuumized for 30 min, and the mother liquor was prepared. The mother liquor was mixed with polyether amine (D230) in a ratio of 1:0.4, stirred again at 500 rpm for 3 min to make the mixture uniform, and vacuumized again for 1 min. In the second step, a mica-graphene/epoxy composite coating was prepared as follows: graphene (graphite powder, natural briquette grade, ~100 nm) was added to 35 wt.% 800# mica/epoxy coating in different contents, stirred at a rotational speed of 1200 r/min for 1 h, dispersed by ultrasonic for 30 min. and vacuumized for 15 min. The coatings had graphene contents of 0.2 wt.%, 0.5 wt.%, 0.7 wt.%, and 1.0 wt.%, and were labeled as TWG-0.2, TWG-0.5, TWG-0.7, and TWG-1.0, respectively. The thickness of the coatings was maintained at 200 ± 10 µm. Please see Paragraphs 2 in Page 3.
- In the case of water absorption rate determination, the immersion medium is not stated. I suppose it is the same 10wt.% NaCl solution.
Response: Thanks for your prompting. The test solution was 10 wt% NaCl, which has been added in the manuscript. Please see Paragraphs 4 Last line in Page 3 and Table 1.
- Corrosion current density is usually denoted using icorr, while the corrosion current is indicated with Icorr.
Response: The errors have been revised, Please see section 3.2 Polarization curve in Page 6 - Page 8.
- Did the authors measure the electrical conductivity of such coatings?
Response: Polarization curve of the coating was measured on the basis of certain electrical conductivity of the composite coatings, so the conductivity of the coatings was not measured individually.
- Why did the authors not investigate the bulk epoxy coating without mica or graphene, as reference material in similar testing conditions?
Response: The anti-corrosion property of the bulk epoxy coating without mica or graphene was limited in oil and gas fields. Therefore, at present, the addition of mica to EP as a filler does not considerably improve the corrosion resistance of the coating, and the infiltration of the coating is not effectively reduced. Furthermore, the coating will have stress concentration during curing, resulting in internal defects [9, 21-23]. However, there are fewer reports on the use of graphene in combination with other fillers in epoxy resins. The purpose of this manuscript is to analyze the synergistic effect of the different filler added to the epoxy coating.
- In the abstract it is stated that the surface characteristics were observed by scanning electron microscopy. The current manuscript does not provide any SEM micrographs, although some aspects about pores and graphene distribution in the coating are discussed, without being sustained. (e.g. …excessive graphene, which were randomly distributed in the coating).
Response: “the surface characteristics were observed by scanning electron microscopy” has been deleted from abstract section..
The morphology of the coating can not characterize the random distribution of graphene, so the statement mentioned above has been deleted.
- Figure 2 (c) has an inferior quality compared to the other 2 related figures (2a and 2b). Furthermore, in figure 2a, the logarithmic axis has different values (ranging from 1014 to 108) compared to the vales reported in Table 2, where there is a minus sign at the exponential factor. Please check those values again.
Response: The quality of Figure 2 (c) has been enhanced, and the order of magnitudes has been also revised. Please Fig. 2(c) in Page 6 and Table 2 in Page 5.
- Moreover, I find figure 3 redundant, as it has a low quality and it does not offer any significant information.
Response: Thanks for your suggestions. Figure 3 mainly describes the macromorphology of the coating before and after immersion at high temperature and high pressure, especially the bubbling characteristics on the surface of the coating, suggest keeping the figure.
- The adhesion test should be sustained with some optical/SEM micrographs.
Additionally, I recommend an article on similar investigations, that can be consulted: Cai, Y.; Meng, F.; Liu, L.;Liu, R.; Cui, Y.; Zheng, H.; Wang, F. The Effect of the Modification of Mica by High-Temperature Mechanochemistry on the Anticorrosion Performance of Epoxy Coatings. Polymers 2021, 13, 378.
Response: Thanks for your recommendation, the quality of the published paper is high. The adhesion test was carried out at the same method, and the detaching status images of TWG-0.7 samples are as follows, which has been added. Please see Fig. 6 in Page 11.

Reviewer 2 Report
This study prepared a new mica-graphene/epoxy composite coating with different contents of graphene to improve the resistance of tubing in the corrosive medium. There are several technical problems need to be addressed. A major revision of the manuscript is required before acceptance for publication.
1. Your conclusions need to be conclusions not restatements of results and a more thorough comparison with existing knowledge is needed. At the moment this reads like a technical report not a scientific paper.
2. The quality (resolution) of figures 2-6 is so poor that results can't be read from the provided figures. High resolution figures are required to replace the current figures.
3. Chapter 4, paragraph 4 - 0.7 wt% is space missing.
4. Chapter 3.2 - 35wt% is space missing.
5. Here are several papers strongly related to your manuscript. Authors may consider quoting.
1) Effects of Chitin Nanocrystals on Coverage of Coating Layers and Water Retention of Coating Color
2) Synthesis of lignin-poly(N-methylaniline)-reduced graphene oxide hydrogel for organic dye and lead ions removal
3) Synthesis and Application of Granular Activated Carbon from Biomass Waste Materials for Water Treatment: A Review
6. Table 2 and Table 5 - The numbers in the last column need superscripts, such as 5.306×107.
Author Response
Reviewer 2:
- Your conclusions need to be conclusions not restatements of results and a more thorough comparison with existing knowledge is needed. At the moment this reads like a technical report not a scientific paper.
Response: Thanks for your review and suggestions, the conclusions have been rewritten, and the relevant literatures were cited to compare the property of the coating, and state the anti-corrosion mechanism. Such as the self-corrosion current density, self-corrosion potential, the resistance Rct etc were compared, and some literatures was cited.
[29] Yang Y.; Gao Y.; Wang X.; An H.; Liang S.; Wang R.; Li N.; Sun Z.; Xiao J.; Zhao X. Preparation and properties of a self-crosslinking styrene acrylic emulsion using amino-functional graphene oxide as a crosslinking agent and anti-corrosion filler. J. Mater. Res. Technol. 2021, https://doi.org/10.1016/j.jmrt.2021.12.114.
[30] Wang X.; Li Y.; Li C.; Zhang X.G.; Lin D.; Xu F.;, Zhu Y.J.; Wang H.Y.; Gong J.L.; Wang T. Highly orientated graphene/epoxy coating with exceptional anti-corrosion performance for harsh oxygen environments. Corros. Sci. 2020, 176, 109049.
Please see Section 3.1 and 3.2, and reference in Pages 4-8 and pages 15-18.
- The quality (resolution) of figures 2-6 is so poor that results can't be read from the provided figures. High resolution figures are required to replace the current figures.
Response: Thanks for your advices, the high resolution figures was used to replace the current figures.
- Chapter 4, paragraph 4 - 0.7 wt% is space missing.
Response: The error has been revised.
- Chapter 3.2 - 35wt% is space missing.
Response: The error has been revised.
- Here are several papers strongly related to your manuscript. Authors may consider quoting.
1) Effects of Chitin Nanocrystals on Coverage of Coating Layers and Water Retention of Coating Color; 2) Synthesis of lignin-poly(N-methylaniline)-reduced graphene oxide hydrogel for organic dye and lead ions removal;3) Synthesis and Application of Granular Activated Carbon from Biomass Waste Materials for Water Treatment: A Review
Response: Thanks for your recommendation. The three papers have been quoted in this manuscript. Please see reference 9, 23, 23, 46, 55.
- Table 2 and Table 5 - The numbers in the last column need superscripts, such as 5.306×107.
Response: The errors have been revised. Please see Table 2 in page 5, Table 5 in Page 10, and the corresponding content in the manuscript.

Reviewer 3 Report
The paper entitled “Corrosion Resistance Mechanism of Mica-Graphene / Epoxy Composite Coating in CO2-Cl- System” focuses on the effects of different contents of graphene on the corrosion resistance of the composite coatings. The coating performance was evaluated by electrochemical techniques and high temperature and high-pressure tests. Тhe introduction refers to the aim of the study, the experimental part is consistently revealed and explained while the results are understandably submitted and sufficiently illustrated. The conclusion summarizes the aforementioned results. In my opinion, the paper should be interesting from a scientific and practical point of view.
I would like to recommend the publication of paper publication after some changes concerning the following issues:
1. In section 2, please indicate the chemical composition of the substrate material;
2. It should be mentioned in the experimental part how the coating is obtained. How does the different viscosity dew to the various graphene concentrations influence the thickness of the coatings?
3. It is not clear for the readers why have the authors exactly chosen the fourth different contents (0.2 wt.%, 0.5 wt.%, 0.7 wt.%, and 1.0 wt.%) of graphene. Additionally, why is the graphene added to a mixture with a concentration of 35 wt.% 800# mica/epoxy;
4. It would be better for the authors to reveal the microstructure and thicknesses of the examined coatings before disclosing the electrochemical performance of the coated material. This will help for a better explanation of the corrosion resistance mechanism.
5. It is recommendable the authors add not only TWG as a control sample but also uncoated substrate material to appreciate the overall influence of the coating on improving the metal pipes performance;
6. Table 3 should be re-located to section 2.
7. The macro-images shown in Figure 3 are unclear. Please use other magnification or phase contrast to underline the occurring changes. SEM could also be a suitable tool for that;
8. Please, improve the dpi quality of Figures 2, 3, 4, and 5.
9. The results presented in Fig. 4 seem statistically unreliable. Standard deviation values should be added to the points in the graph. The same for the values shown in Tables 4 and 6.
10. Please, pay attention to the indices and powers in the text, figures, and tables;
- In some places, the English style and grammar have to be improved. Some examples of inadequate style: “While when the addition amount of graphene was 1.0 wt.%, the values of Rc and Rct of the coating decreased, but them always higher than those of the coating without graphene, which is attributed to the fact that the lamellar structure”; “Generally, the lower the corrosion potential is, the smaller the corrosion current is, and the better the corrosion resistance of the matrix is” …and many more….
11. Where applicable, the obtained results can be compared with other similar studies;
Author Response
Reviewer 3:
- In section 2, please indicate the chemical composition of the substrate material;
Response: Thanks for your review and suggestions. The chemical composition of the substrate material has been indicated. Please the Lines 1-2 in Page 3.
The chemical composition of N80 carbon steel (wt.%) is C: 0.3~0.4, Si: 0.2~0.4, Mn: 1.5~1.8, Cr: 0.15~0.2, P: ≤0.15, S: ≤0.15, and Fe balance.
- It should be mentioned in the experimental part how the coating is obtained. How does the different viscosity dew to the various graphene concentrations influence the thickness of the coatings?
Response: The prepare processes of the coating have been mentioned in Section experimental. However, the thickness of the coatings was artificially controlled to 200 mm for the study the influence of different immersion time, besides the various graphene concentrations. Please see Section 2.1 in Page 3.
- It is not clear for the readers why have the authors exactly chosen the fourth different contents (0.2 wt.%, 0.5 wt.%, 0.7 wt.%, and 1.0 wt.%) of graphene. Additionally, why is the graphene added to a mixture with a concentration of 35 wt.% 800# mica/epoxy;
Response: The previous research found that 35 wt.% 800# mica/epoxy has good corrosion resistance and mechanical property. The purpose of this manuscript is to analyze the synergistic effect of graphene along with mica added to the epoxy coating on the corrosion resistance.
In order to investigate the corrosion resistance of 35 wt.% 800# mica/epoxy resin with different contents of graphene, the authors selected (0.2 wt.%, 0.5 wt.%, 0.7 wt.% and 1.0 wt.%) graphene as fillers. And to accurately locate the content value, the selected intermediate interval was relatively small based on reference 29.
- It would be better for the authors to reveal the microstructure and thicknesses of the examined coatings before disclosing the electrochemical performance of the coated material. This will help for a better explanation of the corrosion resistance mechanism.
Response: The coatings used in this study were 200±10 µm in thickness, which has been added to the manuscript.
- It is recommendable the authors add not only TWG as a control sample but also uncoated substrate material to appreciate the overall influence of the coating on improving the metal pipes performance;
Response: Thanks for your suggestions. This purpose of this manuscript is to analyze the influence of the grapheme content on corrosion resistance of the composite coating. So the corrosion behavior of the metallic matrix (N80 carbon steel) was not stated in the manuscript.
- Table 3 should be re-located to section 2.
Response: The errors in Table 3 have been revised, including the numbers in the last column. Please see Table 3 in Page 7.
- The macro-images shown in Figure 3 are unclear. Please use other magnification or phase contrast to underline the occurring changes. SEM could also be a suitable tool for that;
Response: The quality of images shown in the manuscript has been adjusted, and Figure 6 has been added to state the surface characteristics. Please see Fig. 3 in Page 8 and Fig. 7 in Page 13.
- Please, improve the dpi quality of Figures 2, 3, 4, and 5.
Response: The dpi quality of Figures 2, 3, 4, and 5 has been improved.
- The results presented in Fig. 4 seem statistically unreliable. Standard deviation values should be added to the points in the graph. The same for the values shown in Tables 4 and 6.
Response: Three parallel samples were measured in each group to ensure the reliability. According to your advices, the standard deviation values have been added to the points in the graph, Tables 4 in Page 9 and Table 6 in Page 11.
- Please, pay attention to the indices and powers in the text, figures, and tables; In some places, the English style and grammar have to be improved. Some examples of inadequate style: “While when the addition amount of graphene was 1.0 wt.%, the values of Rc and Rct of the coating decreased, but them always higher than those of the coating without graphene, which is attributed to the fact that the lamellar structure”; “Generally, the lower the corrosion potential is, the smaller the corrosion current is, and the better the corrosion resistance of the matrix is” …and many more….
Response: English style and grammar has been edited by the editing services listed at https://www.mdpi.com/authors/english, Please see the editing certificate.
- Where applicable, the obtained results can be compared with other similar studies;
Response: Thanks for your advices, other similar studies have been cited to compared with the obtained results. Such as the self-corrosion current density, self-corrosion, the resistance Rct etc were compared, and some literatures was cited.
[29] Yang Y.; Gao Y.; Wang X.; An H.; Liang S.; Wang R.; Li N.; Sun Z.; Xiao J.; Zhao X. Preparation and properties of a self-crosslinking styrene acrylic emulsion using amino-functional graphene oxide as a crosslinking agent and anti-corrosion filler. J. Mater. Res. Technol. 2021, https://doi.org/10.1016/j.jmrt.2021.12.114.
[30] Wang X.; Li Y.; Li C.; Zhang X.G.; Lin D.; Xu F.;, Zhu Y.J.; Wang H.Y.; Gong J.L.; Wang T. Highly orientated graphene/epoxy coating with exceptional anti-corrosion performance for harsh oxygen environments. Corros. Sci. 2020, 176, 109049.
Please see Section 3.1 and 3.2, and reference in Pages 4-8 and Pages 15-18.

Round 2
Reviewer 1 Report
The quality of Fig. 2, Fig. 4 and Fig. 5 still must be improved.
Author Response
Thanks for your review and advices again. In the revised manuscript, the DPI resolution of Fig. 2, Fig. 4 and Fig. 5 has been enhanced to 1200, the x-coordinate and the y-coordinate of Figures have been adjusted, and the labels and lines have also been bolded. Please see Fig. 2 in Page 6, Fig. 4 in Page 9 and Fig. 5 in Page 11.

Reviewer 2 Report
It's OK to be published.
Author Response
Thanks for your review and recognition again.
Reviewer 3 Report
Тhe authors have substantially improved the manuscript. However, the quality of some of the figures is still low.
Author Response
Thanks for your review and advices again. In the revised manuscript, the DPI resolution ofFig.1, Fig. 2, Fig. 4 and Fig. 5 has been enhanced to 1200, the x-coordinate and the y-coordinate of Figures have been adjusted, and the labels and lines have also been bolded. Please see Fig.1 in Page 5, Fig. 2 in Page 6, Fig. 4 in Page 9 and Fig. 5 in Page 11. In addition, the contrast of Fig. 3, Fig.6 and Fig.7 has been enhanced.Please see Fig. 3 in Page 9, Fig.6 in Page 12 and Fig.7 in Page 14.
